# ChemGraph as an agentic framework for computational chemistry workflows
**Thang D. Pham** [1] ✉, **Aditya Tanikanti** [2] **& Murat Keçeli** [1] ✉

Atomistic simulations are essential in chemistry and materials science but remain challenging to run due to the expert knowledge required for the setup, execution, and validation stages of these calculations. We present ChemGraph, an agentic framework powered by artificial intelligence and state-of-the-art simulation tools to streamline and automate computational chemistry and materials science workflows. ChemGraph leverages graph neural network-based foundation models for accurate yet computationally efficient calculations and large language models (LLMs) for natural language understanding, task planning, and scientific reasoning to provide an intuitive and interactive interface. We evaluate ChemGraph across 13 benchmark tasks and demonstrate that smaller LLMs (GPT-4o-mini, Claude-3.5-haiku, Qwen-2.5-14B) perform well on simple workflows, while more complex tasks benefit from using larger models. Importantly, we show that decomposing complex tasks into smaller subtasks through a multi-agent framework enables GPT-4o to reach perfect accuracy and smaller LLMs to match or exceed single-agent GPT-4o's performance in these benchmarks.

Atomistic simulations play an important role in chemistry and materials science[1–4], enabling the design of new and improved catalysts[5,6], energy storage materials[1,7,8], and accelerating drug discovery[9,10]. Techniques such as density functional theory (DFT), coupled cluster (CC) methods, molecular dynamics (MD), and Monte Carlo (MC) simulations have been widely adopted to predict molecular and material properties, explore reaction mechanisms, and optimize performance at the atomic level[11–13]. Recent advances in machine learning, particularly the development of graph neural networks (GNNs) and foundation models for molecules and materials, have enabled highly accurate and scalable alternatives to traditional quantum mechanical methods[14–17]. These models are typically trained on large datasets generated from DFT calculations, allowing them to produce DFT-level accuracy at a fraction of the computational cost. Furthermore, their fast inference times enable interactive and real-time molecular simulations, which opens new possibilities for user-driven exploration in chemistry and materials research.

Running molecular simulations efficiently remains a complex and time-consuming task, often requiring specialized expertise and manual effort across the workflow. Researchers must carefully define system parameters, select appropriate computational methods, and generate input files, all tailored to their specific needs. This process is further complicated by the diversity of simulation software, each with its own input syntax, programming language, and interfaces. Even a minor mistake in the input configuration can lead to incorrect results, unnecessary computational expenses,

or failures. To address these challenges, Python libraries such as the Atomic Simulation Environment (ASE)[18] and QCEngine[19] provide a unified interface and act as universal executors for many simulation software packages. Additionally, open databases like the Materials Project[20] and Open Molecules 2025 (OMol25)[21], which publish both simulation inputs and outputs, promote reproducibility and help users adopt widely accepted parameters and practices across a broad range of molecular simulation tasks.

Recent advancements in artificial intelligence (AI), particularly in large language models (LLMs), have opened new possibilities for automating scientific research[22]. LLMs have demonstrated exceptional capabilities in natural language understanding, reasoning, and task execution, making them well-suited for guiding complex workflows[23,24]. Indeed, several LLM-based agents (assistants) have been developed to help users in various chemistry-related tasks[25–29]. Bran et al. developed ChemCrow, a large language model (LLM)-powered chemistry agent capable of executing chemical synthesis for a variety of molecules[25]. Additionally, ChemCrow demonstrates a human-AI collaboration framework that enhances the LLM agent's ability to discover novel molecules. McNaughton et al. developed Chemistry Agent Connecting Tool Usage to Science (CACTUS), an LLM-based agent integrated with cheminformatics tools that can assist researchers in tasks such as molecular property prediction, similarity searching, and drug-likeness assessment[26]. Recently, White and colleagues introduced MDCrow, an LLM agent assistant that can perform molecular dynamics (MD) workflows[28]. Aspuru-Guzik and

[1]Computational Science (CPS) Division, Argonne National Laboratory, Lemont, IL, USA. [2]Argonne Leadership Computing Facility (ALCF) Division, Argonne National Laboratory, Lemont, IL, USA. ✉e-mail: tpham@anl.gov; keceli@anl.gov

colleagues released El Agente, an LLM agent for quantum chemistry calculations[29].

In this work, we introduce ChemGraph[30], a LLM-powered agent system designed to perform molecular simulation workflows in computational chemistry. ChemGraph integrates natural language processing with simulation tools to perform a series of tasks ranging from SMILES string and molecular structure generation to geometry optimization, vibrational analysis and thermochemistry calculations. ChemGraph allows users to utilize a wide range of molecular simulation methods, including semi-empirical methods, ML potentials, and DFT. Our evaluation demonstrates that for simpler tasks requiring only a few tool calls, smaller LLMs such as GPT-4o-mini and Claude-3.5-haiku achieve relatively high accuracy and consistency. However, as the task complexity increases, their performance declines significantly, while a large model (GPT-4o) maintains strong performance. By strategically decomposing complex tasks into smaller, manageable subtasks, we improve the performance of ChemGraph even when using smaller models, achieving results that are comparable to, and in some cases even surpass, those of GPT-4o.

By abstracting away low-level coding and tool-specific configurations, ChemGraph allows users to perform molecular simulations using intuitive prompts. This approach not only lowers the barriers for computational chemistry research but also provides automation that can enhance high-throughput workflows.

## Results
### ChemGraph workflow demonstration
Figure 1 demonstrates how ChemGraph invokes tool calls and coordinates their outputs to complete a computational chemistry task. In this example, the user asks ChemGraph to calculate the reaction enthalpy for methane combustion reaction at 400 K using the GFN2-xTB method. ChemGraph

(with GPT-4o-mini) first converts the chemical names into SMILES strings for each molecule in the reaction (tool calls 1 to 4). It then generates atomic coordinates in the AtomsData data structure based on these SMILES strings (tool calls 5 to 8). Finally, ChemGraph runs thermodynamic calculations using the generated coordinates and user-specified parameters (e.g., calculator, temperature). This example demonstrates how ChemGraph can autonomously use tools and perform multiple intermediate steps to reach the final enthalpy change of the reaction.

### Single-agent evaluations
We evaluated the single-agent ChemGraph's performance on 13 experiments for three LLMs, GPT-4o-mini, Claude-3.5-haiku, and Qwen-2.5-14B. For GPT-4o, we only evaluated its performance on the last two problems, *react2enthalpy* and *react2gibbs*. All experiments were performed on a compute node of the Aurora supercomputer at Argonne Leadership Computing Facility (ALCF). Our timing data showed that the average additional overhead introduced by LLMs was typically less than one minute per instance (excluding simulation time). Detailed timing results are provided in Supplementary Figs. 1 and 2.

Figure 2 shows the accuracy of different models for different tasks. The accuracies of the last two experiments, *react2enthalpy* and *react2gibbs*, are averaged over three independent runs. The task complexity, measured by the number of tool calls, is illustrated in Fig. 3.

The first six experiments (*name2smi* to *name2file*) focus on cheminformatics and molecular simulation tasks using molecule names and other parameters as input. Both Claude-3.5-haiku and GPT-4o-mini demonstrated strong and consistent performance, achieving over 80% accuracy in each task. Qwen-2.5-14B showed more variability in performance, with an accuracy as low as 60% for the *name2gibbs* and *name2-vib* tasks.

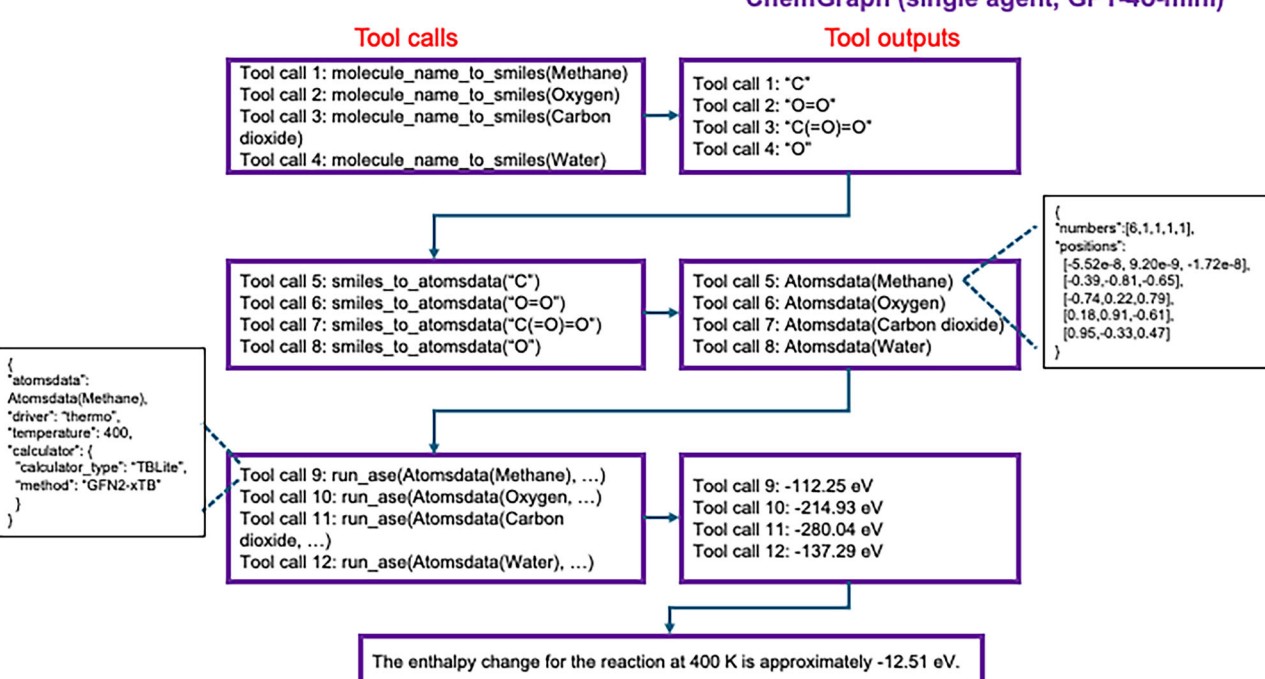

**Fig. 1 | Example of a human-ChemGraph (single-agent) interaction using GPT-4o-mini for a task in *react2enthalpy*.** The user asks ChemGraph to calculate the enthalpy change for the combustion of methane at 400 K using the GFN2-xTB method. The blue box shows the human prompt, while the purple boxes display ChemGraph's outputs, organized into "Tool calls" and "Tool outputs" for easier visualization. For readability, long parameter lists in tool calls and outputs are simplified. Sample AtomsData data structure for methane (tool call 5) and the input parameter for run_ase tool (tool call 9) are shown as dictionaries.

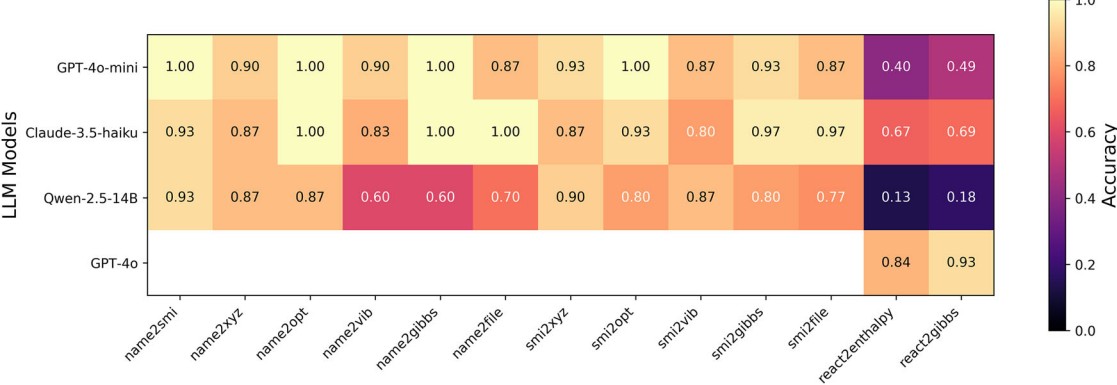

**Fig. 2 | Accuracy heatmap for single-agent ChemGraph.** The heatmap shows averaged accuracy values using different LLMs (GPT-4o-mini, Claude-3.5-haiku and Qwen-2.5-14B) across 360 evaluation instances, grouped into 13 benchmark experiments. GPT-4o was only evaluated on 2 experiments, *react2enthalpy* and *react2gibbs*.

**Fig. 3 | Number of tool calls made by different LLMs across 13 experiments, compared to a baseline representing the number of tool calls required when solved manually by a human.** GPT-4o was evaluated on 2 experiments, *react2enthalpy* and *react2gibbs*. Error bars represent the standard deviation of the number of tool calls per molecule for each experiment.

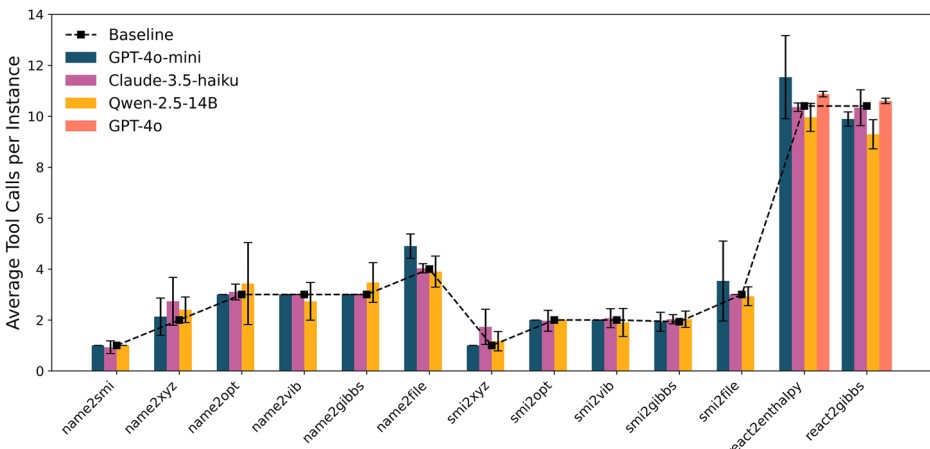

For the *name2smi* task, instead of consistently invoking the *molecule_name_to_smiles* tool, Claude-3.5-haiku sometimes attempted to manually construct the SMILES strings using its internal chemical knowledge or declined to use the tool. Figure 4 illustrates an example with a molecule 1-(2-methylphenyl)-3-(3-methylpyridin-2-yl)urea, where Claude-3.5-haiku declined to invoke the *molecule_name_to_smiles* tool, assuming that the molecule's systematic name would not be compatible with it. Although the model can produce the correct output when explicitly prompted to use the tool, this requires additional user intervention. Because our evaluation focuses strictly on single-turn performance, the responses are still marked incorrectly. The average number of tool calls per molecule for Claude-3.5-haiku for this task, as a result, is the lowest among the three models (Fig. 3). GPT-4o-mini accurately created correct answers for this experiment. Qwen-2.5-14B generally produced accurate tool calls, including both tool names and arguments. However, its errors frequently occurred when extracting the results from the tool outputs.

The next three tasks, *name2xyz*, *name2opt*, and *name2vib*, require the LLMs to not only perform accurate tool calls but also generate correct and structured outputs. These outputs typically include a nested dictionary of atomic coordinates or a list containing both floating-point and complex numbers for vibrational frequency analysis. GPT-4o-mini and Claude-3.5-haiku performed well across these tasks with accuracies above 83%. The most common errors we found across the models are errors in tool arguments and issues during result summarization by the formatter agent, such as missing values or confusion between float and complex number formats in the final report. Interestingly, for the *name2xyz* and *name2opt* tasks, Claude-3.5-haiku often invoked the *save_atomsdata_to_file* tool after

generating the coordinates, which led to a higher number of tool calls than the other two models (Fig. 3).

For the *name2file* task, the user prompt requests a geometry optimization using the mace_mp method and saving the results as an XYZ file. GPT-4o-mini often made an error when invoking the tool calls by passing the method name mace_mp as both the optimizer and calculator, causing initial failure. However, it frequently recovered using feedback from error messages, achieving an accuracy of 87% (Fig. 2) but requiring more tool calls (average of 4.9 versus the expected 4, as shown in Fig. 3).

The next five experiments (*smi2xyz* to *smi2file*) focus on tasks that use a SMILES string as input. Claude-3.5-haiku and GPT-4o-mini maintained strong and consistent performance, both achieving over 83% accuracy (Fig. 2). Qwen-2.5-14B also showed improvement in this set, with a minimum accuracy of 77%. In general, we observe the same common errors and mistakes that also occurred in the first six experiments. These include incorrect tool call arguments and errors when summarizing results, especially when dealing with complex outputs such as nested dictionaries or long lists of mixed floating-point and complex numbers.

In the last two tasks, *react2enthalpy* and *react2gibbs*, we evaluated GPT-4o alongside the small LLMs. To improve the reliability, both accuracy (Fig. 2) and number of tool calls (Fig. 3) metrics for each model were averaged over three independent runs. Qwen-2.5-14B showed the weakest performance on these tasks with accuracies below 20%. GPT-4o-mini achieved moderate results with accuracies of 40% and 49%. Claude-3.5-haiku outperformed other small LLMs with accuracies of 67% and 69%. GPT-4o achieved the highest performance, with an average accuracy above 83% for both tasks. A common error across these tasks was the tendency of

**Human**

Provide the SMILES string corresponding to this molecule: 1-(2-methylphenyl)-3-(3-methylpyridin-2-yl)urea

**ChemGraph**

I'll help you find the SMILES string for the molecule 1-(2-methylphenyl)-3-(3-methylpyridin-2-yl)urea. To do this, I'll use the molecule_name_to_smiles function. However, in this case, the input is a systematic chemical name rather than a common molecule name, so the function might not work directly.

[…]

Would you like me to help you find alternative ways to represent this molecule?

**Human**

You should use molecule_name_to_smiles tool to obtain the SMILES string

**ChemGraph**

I apologize for my previous hesitation. You're right that I should attempt to use the molecule_name_to_smiles tool. However, the tool is designed for common molecule names, and the given name is a complex systematic name. Let me try to use the tool with the full systematic name

Tool Call:
        Tool name: molecule_name_to_smiles
        Args: 1-(2-methylphenyl)-3-(3-methylpyridin-2-yl)urea

Tool call output: "CC1=C(N=CC=C1)NC(=O)NC2=CC=CC=C2C"

Based on the tool output, the SMILES string for 1-(2-methylphenyl)-3-(3-methylpyridin-2-yl)urea is: CC1=C(N=CC=C1)NC(=O)NC2=CC=CC=C2C
[…]

**Fig. 4 | Example multi-turn interaction with the ChemGraph agent in the *name2smi* task.** The LLM initially failed to invoke the appropriate tool. After being provided with human feedback, it invoked the correct tools and reported the correct answer. For clarity, parts of the conversation have been omitted due to length, indicated by […].

LLMs, particularly the smaller models, to confuse molecular properties (e.g., atomic numbers, coordinates, energies) when the context window became crowded with large amounts of data. A detailed example of such confusion is provided in Supplementary Fig. 3.

## Multi-agent evaluations

In the previous experiments, LLMs generally performed well on tasks requiring up to four tool calls. However, their performance declined significantly in the final two experiments. As the number of tool calls and the amount of inputs and outputs increase, the LLM cannot retain relevant information, leading to incorrect tool calls, hallucinations, or data extraction failures. Reducing the context size for LLMs by including only essential information can mitigate this issue. For instance, when calculating the reaction enthalpy, the LLM only needs the enthalpies of formation for each species, not their full atomic coordinates or vibrational frequencies.

To address these challenges, we implemented and evaluated a multi-agent version of ChemGraph, where tasks were distributed among different agents. An example of a human-ChemGraph (multi-agent) interaction is shown in Fig. 5. Similar to the single-agent interaction (Fig. 1), the multi-agent calculates the reaction enthalpy of methane combustion at 400 K using GFN2-xTB. In this workflow, the planner agent decomposes the overall problem into four subtasks of calculating the enthalpy of formation for each molecule. Each subtask is then executed by an executor agent, which invokes a series of tools (molecule_name_to_smiles, smiles_to_atomsdata, run_ase) to perform the calculations. The resulting enthalpies of formation are passed to the aggregator agent, which combines them with the original prompt and the planner's outputs to generate the final reaction enthalpy.

We evaluated the performance of the multi-agent ChemGraph (Fig. 6) using GPT-4o-mini, Claude-3.5-haiku, Qwen-2.5-14B, and GPT-4o. We compared the results against the single-agent performance shown in Fig. 2. The multi-agent system led to significant improvements in accuracy across the models. For the *react2enthalpy* task, GPT-4o-mini improved from 40% to 87%, and Claude-3.5-haiku from 67% to 87%, each value averaged across three independent runs. Notably, both models surpassed the single-agent GPT-4o baseline of 83%. Similar trends were observed in the last task, *react2gibbs*, in which GPT-4o-mini improved from 49% to 87%, and Claude-3.5-haiku from 69% to 93%. In contrast, Qwen-2.5-14B showed only modest gains, primarily due to frequent tool call errors, which limited the aggregator agent's ability to produce an accurate answer. Finally, GPT-4o in the multi-agent setting achieved perfect accuracy, solving both tasks (react2enthalpy and react2gibbs) with 100% success. Identifying the breaking point for GPT-4o would require testing on a new tier of even more complex workflows, but such experiments are currently infeasible due to the significant token costs. We plan to conduct a more systematic and large-scale benchmark with additional tasks and models in a future study.

Our multi-agent design isolates subtasks for each executor agent, keeping its context window short and focused on a single task (calculating properties of a molecule). This prevents confusion, such as coordinate mix-ups observed in the single-agent case (Fig. S1), by reducing the amount of irrelevant information each agent must process. These results demonstrate that decomposing complex workflows into smaller, focused subtasks can substantially enhance LLM performance in molecular simulation workflows.

**Human**

You are given a chemical reaction: 1 (Methane) + 2 (Oxygen) -> 1 (Carbon dioxide) + 2 (Water). Calculate the enthalpy change for this reaction using GFN2-xTB at 400K.

**ChemGraph (multi-agent, GPT-4o-mini)**

**Planner agent**

Task 1: Calculate the enthalpy of formation of methane (CH4) using GFN2-xTB at 400K

Task 2: Calculate the enthalpy of formation of oxygen (O2) using GFN2-xTB at 400K

Task 3: Calculate the enthalpy of formation of carbon dioxide (CO2) using GFN2-xTB at 400K

Task 4: Calculate the enthalpy of formation of water (H2O) using GFN2-xTB at 400K

**Executor agents**

Executor 1

Executor 2

Executor 3

Executor 4

**Tool calls**

Tool call 1: molecule_name_to_smiles(Methane)
Tool call 2: smiles_to_atomsdata("C")
Tool call 3:run_ase(Atomsdata(Methane), driver=thermo, temperature=400, "calculator"={"calculator_type":"TBLite", "method": GFN2-xTB} )

**Tool outputs**

The thermochemical calculation for methane (CH4) using the GFN2-xTB method at 400K has been successfully completed [...] **Enthalpy of Formation**: -112.2512 eV [...]

[...] **Enthalpy of Formation**: -214.93 eV [...]

[...] **Enthalpy of Formation**: -280.0425 eV [...]

[...] **Enthalpy of Formation**: -137.2903 eV [...]

**Aggregator agent**

[...] Thus, the enthalpy change (\u0394H) for the reaction is: Delta H = -12.5119 eV

**Fig. 5 | Example of a human-ChemGraph (multi-agent) interaction using GPT-4o-mini for a task in *react2enthalpy*.** The user asks ChemGraph to calculate the enthalpy change for the combustion of methane at 400 K using the GFN2-xTB method. The blue box shows the human prompt, while the purple boxes display ChemGraph's outputs, organized into "Planner agent", "Executor agents", "Tool calls", "Tool outputs", and "Aggregator agent" for easier visualization. For readability, LLMs' messages and long parameter lists in tool calls and outputs are truncated or indicated as "[...]". The detailed tool calls are only shown for Executor 1 for visualization.

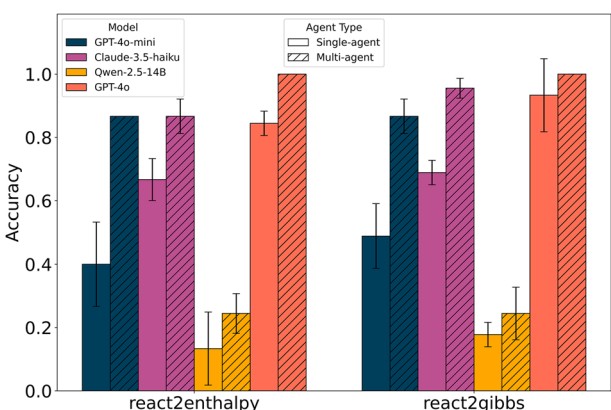

**Fig. 6 | The average accuracy of multi-agent and single-agent ChemGraph using different LLMs for *react2enthalpy* and *react2gibbs* tasks.** Error bars represent the standard deviation over three independent runs.

## Key features of ChemGraph

Recently, several other agentic workflows for materials science and computational chemistry have been introduced. MatAgent[31] by Lv et al. is a single-agent system designed to perform material property retrieval and prediction, relying on PWDFT[32] with GPT-4 as the core reasoning engine. Another framework by Takahara et al. with the same name, MatAgent[33], relies on tool-augmented LLM reasoning (tested with GPT-4o and o3-mini) to plan and justify inorganic material compositions using short/long-term memory, a periodic-table helper, and a mined knowledge base, while coordinating a diffusion-based structure generator and a GNN property predictor to iteratively steer designs toward target values. El Agente[29] is a multi-agent system designed to accelerate computational chemistry tasks such as geometry optimization and property predictions using ORCA[34,35], and xTB[36]. El Agente was evaluated with multiple university-level exercises and tested on two case studies involving solvent effects on IR spectra and modeling of lanthanoid complexes, using GPT-4.1 and Claude-3.7-Sonnet as its reasoning engines. DREAMS (DFT-based Research Engine for Agentic Materials Screening)[37] adopts a multi-agent system for material discovery that utilizes Claude-3.7-Sonnet. DREAMS was benchmarked with three problems, lattice constant prediction, CO adsorption on Pt(111) surfaces, and determination of exchange-correlation functional uncertainty. Aitomia[38] is a multi-agent system capable of performing computational tasks using a wide range of tools, from quantum mechanical methods to machine learning approaches, with DeepSeek-V3 as the core engine and many foundation models supported through the MLatom package.

The rapid growth of agentic workflows highlights the community's interest in accelerating computational chemistry and materials discovery with these AI-based tools. Within this evolving landscape, our goal is to highlight the key features of ChemGraph that contribute to this effort, without asserting exclusivity or novelty, but rather to contextualize our design choices and their impact.

1. **Flexibility and Modularity**. ChemGraph leverages ASE[18] calculators, which enable integration of a wide range of simulation packages. Many modern simulation codes, from semi-empirical approaches like tight-binding, foundation models such as MACE[14,15] and the Universal Models for Atoms (UMA)[16], to ab initio methods as implemented in NWChem and ORCA, can be accessed through ASE calculators. This common layer allows ChemGraph to operate consistently across

different simulation backends without changing its core logic. This modularity not only simplifies implementation but also enables rapid benchmarking and method substitution, depending on the user's desired trade-off between accuracy and computational cost.

2. **Open-Source Framework with Extensive Benchmarking**. ChemGraph is fully open source and includes a publicly available benchmark suite comprising 360 experiments across 13 representative tasks in computational chemistry, from cheminformatics to thermodynamic property prediction. This extensive evaluation across different LLMs provides a robust performance baseline for future work and facilitates transparent comparisons and reproducibility. The open-source nature of ChemGraph also invites community contributions and supports long-term maintainability and innovation.

3. **Cost-Efficient Multi-Agent Design with Small LLMs**. While many agentic workflows rely on large LLMs, ChemGraph emphasizes cost efficiency and scalability by supporting small but capable LLMs such as GPT-4o-mini, Claude-3.5-haiku, and Qwen-2.5-14B. We demonstrate that these models, when used in a multi-agent design, can achieve performance comparable to GPT-4o on a variety of tasks, while offering significantly lower inference costs. This makes ChemGraph practical for real-world scientific applications where cost, latency, and resource constraints are critical considerations. This observation aligns with recent work suggesting that small LLMs are particularly well-suited for agentic AI applications[39].

## Limitations, safety, and future directions

ChemGraph is the initial release of our LLM agent system for computational chemistry and materials science workflows. As the LLM ecosystem advances, we plan to expand ChemGraph's capabilities by integrating new tools, enhancing agent design, and extending compatibility to a broader range of software platforms. A key goal of ChemGraph is to support the execution of large-scale or long-running simulations. To that end, we are actively working on enabling automated generation of job submission scripts and deployment on high-performance computing (HPC) systems. This will empower users to efficiently manage large simulation campaigns across distributed environments.

Safety and reliability are also central to ChemGraph's design. LLM agents can sometimes behave unpredictably or make incorrect decisions. To mitigate this, we provide ChemGraph within a Docker container that ensures controlled execution environments, enforces safe dependencies, and limits unintended interactions with the host system or external resources.

ChemGraph shows strong tool-calling performance across our benchmark tasks, though none of the four evaluated LLMs achieved perfect accuracy, apart from the multi-agent benchmark with GPT-4o. Similar to human-operated workflows, LLMs still occasionally make errors. However, as LLM capabilities and their tool-calling capability continue to improve, we anticipate a corresponding boost in ChemGraph's effectiveness and robustness.

In addition to these directions, we view ChemGraph as complementary to existing graphical user interfaces (GUIs) for computational chemistry packages. GUIs are valuable for lowering the barrier to entry for standardized tasks, yet their scope is often limited, and deploying them effectively on high-performance computing (HPC) systems for large or highly customized workflows can be challenging. In contrast, ChemGraph's LLM-based agents enable researchers to create, modify, and execute workflows dynamically through natural language, offering a level of flexibility that fixed GUIs cannot easily provide, whether in local or HPC environments. This adaptability, together with the ability of LLMs to explain intermediate steps and recover from errors, positions ChemGraph not as a replacement for GUIs but rather as a collaborative layer that extends and enhances their functionality. To underscore this vision, we provide an experimental GUI for ChemGraph in our codebase. This application demonstrates how a natural language interface can work alongside visual components to create a

more powerful and intuitive scientific workflow tool, representing an initial step toward a more synergistic user experience.

Our long-term vision is to make ChemGraph widely accessible to researchers, providing a natural language interface to high-quality simulation tools. One current limitation is the cost associated with using proprietary models. To address this, our results highlight the potential of combining ChemGraph with open-source models like Qwen-2.5-14B, enabling cost-effective execution of complex scientific workflows. We believe the first version of ChemGraph provides a groundwork for systems of intelligent scientific agents that can not only automate simulations but also adapt, reason, and collaborate, paving the way for more efficient and accessible scientific discoveries.

## Conclusions

In this work, we introduced ChemGraph, an LLM-powered agentic framework designed to automate molecular simulation workflows through structured tool calling and reasoning. We evaluated ChemGraph across a diverse set of 13 experiments, ranging from simple molecule-name-to-SMILES conversions to complex thermodynamic property calculations, and demonstrated that state-of-the-art LLMs can achieve high task completion accuracy.

We found that single-agent systems perform reliably on tasks involving a small number of tool calls. However, performance degrades for complex workflows due to context window saturation. To address this, we implemented a multi-agent version of ChemGraph that decomposes complex queries into smaller subtasks across agents. This approach significantly improved accuracy, especially for reaction enthalpy and Gibbs free energy calculations, where GPT-4o achieved perfect accuracy (100%), and small LLMs exhibited substantial gains, even surpassing the single-agent performance of larger models.

Another key strength of ChemGraph lies in its modular design and compatibility with ASE-style calculators, enabling integration of a wide range of simulation backends. We demonstrated ChemGraph's flexibility by executing workflows with DFT, tight-binding, and state-of-the-art machine learning potentials, demonstrating its potential for rapid testing and benchmarking across computational methodologies. By coupling ChemGraph with fast and accurate machine learning potentials, we enable interactive and natural language-driven molecular simulations that support the efficient and accurate exploration of chemical space.

## Methods
### Framework

ChemGraph is implemented with LangGraph[40] and follows the ReAct framework[41]. LangGraph introduces a graph-based execution model designed for more robust multi-agent coordination. A typical LangGraph workflow consists of three components: state, nodes, and edges. The state is a shared data structure representing the system's status. Nodes are Python functions that define the logic of agents, taking the current state as input and returning an updated state. Nodes can be LLMs or functions executing a specific operation. Edges are functions that control the flow of messages, determining which nodes are executed next. Edges can be *directed* or *conditional*. A *directed* edge represents a fixed, one-way flow of messages from one node to another. In contrast, a *conditional* edge introduces flexibility, allowing messages to reach different nodes based on a condition.

Figure 7 illustrates the generalized representation of ChemGraph. First, the LLM agent is provided with a predefined set of tools (defined in the next section). Based on the user prompt, the agent decides which tool to invoke. After each tool call is finished, the LLM is provided with the tool-call result and decides if another tool call is needed. After all tool calls are completed, the message can follow one of the two paths. In the first path, ChemGraph acts like a conventional LLM and returns a human-like response that incorporates results from the tool calls. In the second mode, the messages are forwarded to a second LLM agent responsible for producing a structured output (with a predefined JSON schema) that directly addresses the user's request. This dual-mode design supports both natural language interaction

**Fig. 7 | Overview of ChemGraph.** Solid arrows represent direct edges, indicating the default flow of messages. Dashed arrows represent conditional edges, where message flow depends on specific conditions. The figure lists the libraries used for workflow and cheminformatics, and for simulation backends.

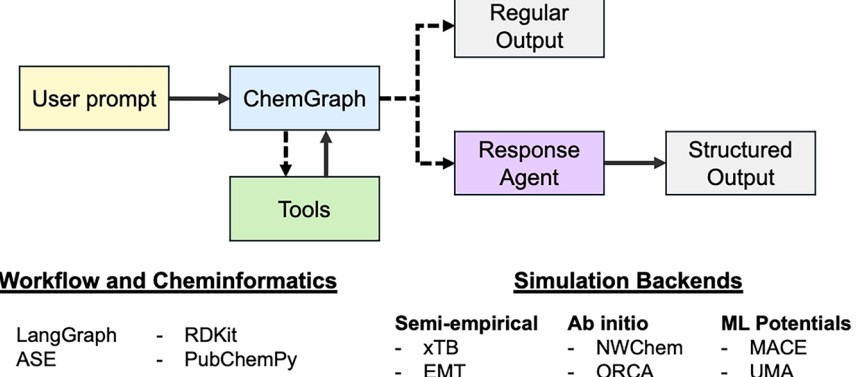

## Table 1 | Tools integrated within ChemGraph, including their names and brief descriptions of their functionalities

| Tool name | Tool Description |
|---|---|
| molecule_name_to_smiles | Convert a molecule name to a SMILES string |
| smiles_to_atomsdata | Convert a SMILES string to an AtomsData object |
| file_to_atomsdata | Convert a file to an AtomsData object |
| run_ase | Run molecular simulation (optimization, vibrational analysis, thermochemistry calculations) via ASE |
| save_atomsdata_to_file | Save an AtomsData object to a file (XYZ, mol, etc.) |
| calculator | Calculate a mathematical expression |

and systematic evaluation, enabling ChemGraph to serve both typical use cases and evaluation workflows.

### Tools

In ChemGraph, tools are implemented as Python functions that wrap widely used libraries such as RDKit[42], Atomic Simulation Environment (ASE)[18], and PubChemPy[43] into structured and agent-compatible interfaces. The tools cover a broad range of functionalities, from basic cheminformatics operations such as converting molecule names to SMILES strings and generating 3D molecular structures, to more advanced simulation workflows using ASE, including geometry optimization, vibrational frequency analysis, and thermochemistry calculations. Several tools use an internal data structure called AtomsData, which is a lightweight wrapper around the ASE Atoms class. It retains key information such as atomic numbers, atomic coordinates, cell parameters, and periodic boundary conditions. The primary purpose of AtomsData is to make atomic structure data serializable and compatible with LangGraph message passing. A summary of the tools utilized and evaluated in this study is presented in Table 1, with detailed specifications of their inputs and outputs available in Supplementary Table 1.

The use of ASE as our core simulation engine enables us to access a broad range of computational chemistry methods. Through its modular calculator interface, ChemGraph can access methodologies ranging from fast, low-cost approximations such as the extended tight-binding (xTB) method[36] and ML potentials like MACE[14,15], to more accurate but computationally intensive approaches, including DFT and post-Hartree-Fock methods via NWChem[44] and ORCA[34,35]. This versatility allows users to balance computational cost and accuracy depending on the task, making ChemGraph suitable for both high-throughput screening and high-accuracy electronic structure calculations.

### Prompt strategy

Prompts play an important role in establishing context and guiding the agent's behavior. In LangGraph's framework, the system prompt defines the available tools, the agent's responsibilities, and the expected structure of outputs. Kumar and colleagues have shown that having a modified prompt to align the agent more with the domain of chemistry can help improve the success of tool calls[26]. In ChemGraph, we experimented with both general-purpose prompts and model-specific variants to optimize performance across different LLMs. These customized prompts included chemistry-specific language, general clarifications on tool input formats, and explicit instructions to avoid common failure modes (e.g., unnecessary tool calls or hallucinated tool outputs).

### Benchmarking and Evaluation

While several agentic benchmarks exist[45,46], to the best of our knowledge, no common benchmark currently exists for agentic evaluation in computational chemistry. As a result, we developed 13 benchmark experiments to evaluate ChemGraph. These experiments assess ChemGraph's ability to utilize six integrated tools, either through single tool calls or chained tool calls, organized into three categories based on the type of user input: (1) molecule names, (2) SMILES strings, and (3) chemical reactions. Table 2 summarizes the 360 independent evaluations conducted across the 13 experiments. In the first 11 experiments, which focus on tasks involving molecule names or SMILES, each requires up to four tool calls or subtasks. In contrast, the final two experiments, centered on computing thermochemical properties of chemical reactions, are significantly more complex, requiring between 9 and 12 tool calls depending on the number of reactants and products. These tasks involve executing thermochemistry calculations for each species individually and creating a final reaction-level property based on previous outputs. The sample human prompt for each task is shown in Supplementary Table 2.

For each experiment, we randomly selected a set of molecules from PubChem[43], setting a constraint on the maximum number of atoms per molecule, as shown in Supplementary Table 2. This constraint ensures that each evaluation instance can be completed within a reasonable time, with stricter cutoffs for more computationally intensive tasks. The *name2opt* and *smi2opt* tasks are exceptions; for these, we used a fixed list of small molecules across all runs because the geometry optimizations were performed using DFT.

We evaluated ChemGraph using a single-turn framework. Single-turn and multi-turn evaluations represent two different approaches to assessing the performance of LLM agents[47]. In a single-turn evaluation, the agentic framework receives a user query and generates a complete response without human intervention. If the LLM makes a mistake in a tool call, it can attempt to self-correct within that single interaction. In contrast, multi-turn evaluation involves iterative human-AI interaction where humans provide corrective feedback when the AI makes mistakes, allowing for correction and refinement of the result through multiple exchanges. While ChemGraph allows multi-turn interaction and human-AI collaboration, multi-turn evaluation presents significant methodological challenges: different LLMs may exhibit different error patterns, leading to divergent correction paths and making it difficult to establish consistent and reproducible benchmarks.

## Table 2 | Summary of benchmark experiments used to evaluate ChemGraph

| Label | Description | Number of subtasks | Number of instances |
|---|---|---|---|
| name2smi | Convert a molecule name to a SMILES string | 1 | 30 |
| name2xyz | Convert a molecule name to XYZ coordinates | 2 | 30 |
| name2opt | Run a geometry optimization given a molecule name | 3 | 30 |
| name2vib | Run a vibrational frequency calculation given a molecule name | 3 | 30 |
| name2gibbs | Calculate the Gibbs free energy given a molecule name at a temperature | 3 | 30 |
| name2file | Run geometry optimization and save the coordinates into a named XYZ file given a molecule name | 4 | 30 |
| smi2xyz | Convert a SMILES string to XYZ coordinates | 1 | 30 |
| smi2opt | Run geometry optimization given a SMILES string | 2 | 30 |
| smi2vib | Run a vibrational frequency calculation given a SMILES string | 2 | 30 |
| smi2gibbs | Calculate the Gibbs free energy of a given SMILES string at a temperature | 2 | 30 |
| smi2file | Run geometry optimization and save the coordinates into a named XYZ file for a given SMILES string | 3 | 30 |
| react2enthalpy | Calculate the reaction enthalpy at a temperature for a given reaction | 9 to 12 | 15 |
| react2gibbs | Calculate the reaction Gibbs free energy at a temperature given a reaction | 9 to 12 | 15 |

Each row includes the experiment label, a brief description, the number of subtasks, and the number of evaluation instances (e.g., number of molecules, SMILES strings, or reactions).

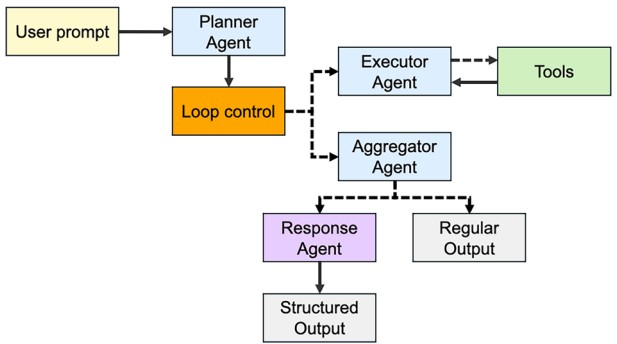

**Fig. 8 | Architecture of the multi-agent ChemGraph.** ChemGraph includes a series of specialized agents (Planner, Executor, and Aggregator), coordinated by loop control logic. Outputs are generated in either regular or structured formats. Dashed arrows represent conditional edges, where message flow depends on specific conditions.

To benchmark ChemGraph, we designed a standardized evaluation procedure with a template-based prompting approach. Each task was defined using a template prompt, with variable components, such as molecule names, SMILES strings, or sets of reactants and products. These components are systematically varied in each instance of each experiment while keeping other components fixed, such as computational methods (e.g., DFT, semi-empirical, ML potentials), conditions (e.g., temperature), and file naming conventions. For every experiment and instance, we created a reference answer in structured JSON format. The reference answer represents how a domain expert would solve the problem based on the available tools, defining both the sequence of tool calls and the expected results, which serve as ground truth for evaluation.

Performance of LLMs was evaluated based on two metrics: (1) the accuracy of the final answer, and (2) the number of tool calls used to reach that answer. To enable efficient and consistent benchmarking, we used ChemGraph's second operational mode, which generates structured outputs. We evaluated the performance of ChemGraph using four LLMs, categorized into open and proprietary models. The open model used was Qwen-2.5-14B, a publicly released version of Alibaba's Qwen series. The proprietary models included GPT-4o-mini and GPT-4o from OpenAI, and Claude-3.5-haiku from Anthropic. We set the temperature of each model to zero to ensure consistency. However, we observe that even at zero temperature, the performance of the LLM still varies across different runs. Qwen-2.5-14B was accessed through the Argonne Leadership Computing

Facility (ALCF) using Globus Compute on ALCF's high-performance computing clusters. Access to the three proprietary models was provided via their respective APIs. Due to the high API cost associated with GPT-4o, its evaluation was limited to the two most complex tasks, *react2enthalpy* and *react2gibbs*.

### Multi-Agent System

While a single LLM agent can handle simple tasks, its performance varies depending on the task complexity, especially when the workflow requires multiple or chained tool calls. In contrast, a human expert can break down a complex workflow into smaller and more manageable subtasks. For instance, consider the simplified *react2enthalpy* task of calculating the reaction enthalpy for a reaction $A + B \rightarrow C$. This problem can be broken down into three subtasks: compute the enthalpy of formation for A, B, and C. Once those values are obtained, the overall reaction enthalpy can be determined.

Following this principle, we designed and evaluated a multi-agent version of ChemGraph for the two most complex tasks in this work, *react2enthalpy* and *react2gibbs*. Figure 8 illustrates the multi-agent system's architecture. Instead of assigning the entire workflow to a single LLM, the multi-agent ChemGraph system comprises three main LLM agents: a planner agent, executor agent(s), and an aggregator agent. The planner agent decomposes the user's request into subtasks and generates a series of prompts, as demonstrated previously. These sub-prompts are passed to a loop controller, which sequentially forwards each prompt to the executor agent(s). Each executor is equipped with the same set of tools as in the single-agent system. After all subtasks are completed, their summarized results are sent to the aggregator agent. The aggregator integrates the original user query, the planner's task decomposition, and summarized outputs from executors to generate the final answer.

### Data availability
Evaluation data and source code for ChemGraph are available on Zenodo: https://zenodo.org/records/17290519.

### Code availability
ChemGraph is available on GitHub: https://github.com/argonne-lcf/ChemGraph. ChemGraph framework described in this work used the following software versions: LangGraph v0.4.7, ASE v3.25.0, NWChem v7.2.3, mace-torch v0.3.13, fairchem-core v2.3.0, RDKit v2024.3.5, TBLite v0.4.0, ORCA v6.0. The evaluations of ChemGraph were performed using the following LLM API versions: gpt-4o-mini-2024-07-18, gpt-4o-2024-08-06, claude-3-5-haiku-20241022. The Qwen-2.5-14B model was accessed via ALCF endpoints: https://github.com/argonne-lcf/inference-endpoints.

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

## Acknowledgements
This research used resources of the Argonne Leadership Computing Facility, a U.S. Department of Energy (DOE) Office of Science user facility at Argonne National Laboratory. This work was supported by the Office of Science, U.S. Department of Energy, under Contract No. DE-AC02-06CH11357. Our work leverages ALCF Inference Endpoints, which provide a robust API for LLM inference on ALCF HPC clusters via Globus Compute. We are thankful to Serkan Altuntaş for his contributions to the user interface of ChemGraph and for insightful discussions on AIOps.

## Author contributions
T.D.P. co-designed the ChemGraph framework with M.K., led the implementation, conducted all experiments, and wrote the first draft of the manuscript. A.T. implemented the containerized environment and deployed the LLM inference servers. M.K. conceptualized and led the project, contributed to software development, and revised the manuscript. All authors edited and approved the final version of the manuscript.

## Competing interests
The authors declare no competing interests.
