## [Transparent Peer Review file · Communications Chemistry]

ChemGraph as an agentic framework for computational chemistry workflows

Corresponding Author: Dr Murat Keceli

Version 0:

Reviewer comments:

Reviewer #1

(Remarks to the Author)

In this work, the authors present "ChemGraph", an agentic AI approach to performing computational chemistry tasks. They show that for simple tasks requiring fewer than four tool calls, both commercial and open-weight models can perform well in a single-agent architecture. However, for more complex workflows which require many tool calls and the consolidation of multiple intermediate results, only the most expensive commercial model (GPT-4o) achieves good accuracy. The authors show that using a multi-agent system can allow smaller (cheaper) commercial models to meet or exceed GPT-4o's performance in these complex workflows.

The methods are clearly presented, and the authors clearly and rigorously motivate the extension from a single- to a multi-agent architecture. The discussion of the results is well organized and clear. For these reasons, I believe that their submission could be of interest to the community of Nature Communications Chemistry.

However, there are a few points that I would like the authors to address to improve the clarity and impact of their work.

1. While the authors clearly demonstrate the limits of the single-agent approach w.r.t. task complexity, it would be good if they could also push the multi-agent system to its complexity limit and show where it lies.
2. The authors understand and explain some of the common mistakes and failure modes of the single-agent approach. However, the discussion of the improvements made by the multi-agent system lacks a similar depth. In particular, showing and discussing an exemplary case where the single-agent fails and the multi-agent succeeds would make a strong addition to the manuscript.
3. It is unclear where the key novelty lies in the ChemGraph approach
 - a) for the single-agent case: w.r.t. the integration of new tools in an existing single-agent approach like ChemCrow or CACTUS
 - b) for the multi-agent case: w.r.t. multi-agent systems like El Agente (Q), which the authors cite, and which is integrated with many of the same tools

Reviewer #2

(Remarks to the Author)

Paper Review: ChemGraph: An Agentic Framework for Computational Chemistry Workflows

1. Recommendation

I believe this work introduces an interesting computational chemistry utility to the wider community, but the conversation around the benchmark is not very compelling and some details about the tools themselves are not very clearly articulated.

I'd accept this with Minor to Major Revisions.

2. Summary of the Paper

This paper highlights a new LLM-based tool: ChemGraph, which is a natural language interface to a set of computational chemistry tools through the LangGraph framework. The paper introduces the tool and discusses benchmarking results for a variety of tasks.

This paper makes no major applied scientific claims about their tool, but they discuss its impact in how it can utilize various LLMs to accomplish tasks to varying degrees of success to allow for a wider access to computational chemistry for those unfamiliar with programming or to accelerate computational chemistry research for those that are familiar.

They also discuss the differences in execution between single and multi-agent configurations of their tool.

Overall this is a paper that highlights the creation of an exciting new tool for the computational chemistry community.

3. Strengths

* Framework

I believe the authors do a good job of highlighting the structure of their agent / multi-agent workflow. It is clear how the tools are being used and the format which they are given.

* Diversity in Models

The model selection for this paper is a strong source of comparison to the current state of models. It is an interesting outlook to compare how certain models perform on similar tasks in a controlled setting like the benchmarks provided here.

* Application

The authors have introduced a tool that I could see serving in a useful way to the community. Lowering the barrier of entry to computational chemistry in this way could see a lot of benefits to computational design of molecules and materials.

* Code and Data Availability

The authors have been very transparent in their included code, data, and supporting information. This is a great mechanism to take, and greatly benefits the overall community with sharing approaches to integrating LLMs with scientific tooling.

4. Major Weaknesses / Areas for Major Revision

* Discussion on the Benchmarks

There was some discussion of the benchmark design but it did not go into much detail about the actual format of the questions asked. There is one example of a multi-turn interaction that gives a typical question, but it would be beneficial for an example to be provided for the major experiments. The benefit of an LLM is the natural language approach, so seeing what natural language is asked is helpful. I know there is some information in the codebase itself and the supporting information, but I believe there should be more in the main text.

* The Benchmark

How were the number of instances determined? Why run some at higher instances than others? 30 instances seems insufficient for a sample size and then getting down to 10 for the more complicated questions where you'd like to see a more robust analysis. I understand with LLMs the API calls can be expensive, but do you believe that you are getting a large enough sample size for those sets of questions?

Also is there a sense of time in your benchmarks? How long does a response take to generate single or multi-agent? The computations are rather uncontrollable, but the time to perform a typical ReAct loop from start to end may be worth highlighting.

* Use of LLMs over standard software methods

I understand the benefit to using LLMs to serve as a natural language interface to individuals who may have little experience in programming, but I'd like to see a stronger reason why LLM-agents help to solve this problem in a better way over just a simple graphical user interface that taps into the same backend python wrappers that you have written for the LLM to utilize.

The discussion provided in the 'future directions' section should focus on the strengths of LLM-based agents over rigid GUI

tools that can be designed to accomplish a similar goal.

5. Minor Weaknesses / Suggestions for Minor Revision

* Grammar

There are a few sections that could use another pass on grammar checking to ensure clarity.

A few I noticed:

** Page 2: "that can perform molecular dynamics (MD) workflow" -> "workflows"

** Page 2: "In this work, we introduced" -> "we introduce"

* Computational Power

You mention where you sourced the LLM's from, either HPC or an API, but for actually running the workflow is there a requirement for compute power? Like is this something only a user of an HPC could run? Some discussion about the hardware used to perform the experiments would be beneficial. To clarify my question a little more, what is running the python code?

* Multi-agent evaluation for GPT-4o

Maybe I missed it, but was there a reason there was no multi-agent run of Gpt-4o on the `react2enthalpy` and `react2gibbs` to compare with the other multi-agents?

Reviewer #3

(Remarks to the Author)

The authors present ChemGraph based on LLM agents for autonomous computational chemistry simulations. They perform an important and honest benchmark of some typical tasks, and provide the data and code. I commend the authors for the nice work. It is a timely contribution, given the current trend of rapidly developing agentic tools for computational chemistry.

I only have a few rather technical comments:

1. Please provide the versions of the software used, i.e., Orca, etc, as they sometimes provide different results. Similar to the above, please provide the date of making benchmarks with some of the LLMs via API (i.e., for models which the authors cannot deploy locally, such as GPT-4 series). They might change with time, and we observed that some of these tools remember the previous requests through the API, if not explicitly turned off, which might influence the evaluations. That is, the evaluations might change if they are to be repeated.

2. It would be interesting to see the time to solution for some of the typical tasks and, possibly, a breakdown of how much of the time is taken by agents and time taken by doing actual computational chemistry calculations.

3. Could the author elaborate more on how their work differs from the concurrent related works, i.e., El Agente (cited preprint, but it is now published in DOI 10.1016/j.matt.2025.102263), MatAgent (not cited: <https://doi.org/10.1002/mgea.70013>), Aitomia (not cited, arXiv:2505.08195), and DREAMS (appeared after ChemGraph's submission, arXiv:2507.14267)?

Version 1:

Reviewer comments:

Reviewer #1

(Remarks to the Author)

I'm satisfied with the changes that the authors have made.

Reviewer #2

(Remarks to the Author)

I believe that the authors have done a satisfactory job in responding to all 3 reviewers comments in a way that enhances the quality of the paper.

General Comments:

In reference to my comments, the inclusion of Figure 3 and Figure 7 provide a much more insightful view into the processes and capabilities of ChemGraph. The added discussion on a typical conversation and highlighting how the single-agent / multi-agents decide on their final response is a great benefit to the overall article.

The enhanced discussion on benchmarking and increase of the # of questions / instances is a good step in the right

direction. I agree with the reasoning the authors have provided for why they chose the number they did, and I think the increase to 360 question-answer pairs satisfies any concern I had over not having a representative dataset to identify weaknesses in LLM responses.

The author's inclusion of the multi-agent GPT-4o run is great to see. The results gained from it's inclusion seem to have greatly supported the strengths of multi-agent task completion for this set of problems.

Points to address:

I believe all the points I had were addressed and no new major issues have appeared from the addition of the new content.

Reviewer #3

(Remarks to the Author)

The revision has improved the manuscript and addressed some of the comments. However, it also introduced some new dubious claims and mistakes.

In the revised versions, the references are not complete; the paper cites 47 papers, while the References only list 35 papers.

In the following paragraph, they mention MatAgent twice but give different references to likely different publications: "MatAgent42 by Lv et al. is a single-agent system MatAgent44 by Takahara et al. relies on tool-augmented LLM reasoning (tested with GPT-4o and o3-mini) ..." Unfortunately, it is not possible to check where the mistake lies as the references 42 and 44 are not provided.

In the distinctive features of ChemGraph, they mention flexibility and modularity due to the ASE use. They also emphasize several times the foundation model support, which they attribute in the response letter as 'unique'. None of this is unique, as Aitomia is based on MLatom, which also enables integration of a wide range of simulation packages akin to ASE, and it also supports a host of foundation models (in fact, dozens of them).

Another distinctive feature was said to be the use of small LLMs, but the previous work on Aitomia also explored them.

They also mention the 'vision' of combining ChemGraph with GUIs, but Aitomia also demonstrated the synergic benefits of combining agentic workflows with GUI...

While the above independent work on Aitomia does not take away from the original contribution by the authors, the above statements must be revised to correctly reflect the current state of the art.

Open Access This Peer Review File is licensed under a Creative Commons Attribution 4.0 International License, which permits use, sharing, adaptation, distribution and reproduction in any medium or format, as long as you give appropriate credit to the original author(s) and the source, provide a link to the Creative Commons license, and indicate if changes were

made.

Reviewer #1 (Remarks to the Author):

In this work, the authors present "ChemGraph", an agentic AI approach to performing computational chemistry tasks.

They show that for simple tasks requiring fewer than four tool calls, both commercial and open-weight models can perform well in a single-agent architecture.

However, for more complex workflows which require many tool calls and the consolidation of multiple intermediate results, only the most expensive commercial model (GPT-4o) achieves good accuracy.

The authors show that using a multi-agent system can allow smaller (cheaper) commercial models to meet or exceed GPT-4o's performance in these complex workflows.

The methods are clearly presented, and the authors clearly and rigorously motivate the extension from a single- to a multi-agent architecture. The discussion of the results is well organized and clear. For these reasons, I believe that their submission could be of interest to the community of Nature Communications Chemistry.

However, there are a few points that I would like the authors to address to improve the clarity and impact of their work.

1. While the authors clearly demonstrate the limits of the single-agent approach w.r.t. task complexity, it would be good if they could also push the multi-agent system to its complexity limit and show where it lies.

Author reply: We thank the reviewer for this suggestion. We agree that it is important to explore the true limits of complexity for both single- and multi-agent systems. Our current results demonstrate these limits for the GPT-4o-mini, Claude-3.5-haiku, and Qwen-2.5-72B models, which are unable to achieve 100% accuracy even within the multi-agent design. GPT-4o is the sole exception, as it did reach perfect accuracy, highlighting both the strength of the multi-agent approach and the capabilities of the most advanced models. While we are eager to identify the breaking point for GPT-4o, the main barrier is financial. The significant token costs of exhaustively testing GPT-4o on an even more complex tier of workflows are not currently feasible. We plan to carry out a more systematic study with additional tasks and models in future work once the necessary funding becomes available. (We currently use our personal accounts for OpenAI and Claude API access.)

We added GPT-4o to the multi-agent tests and revised the discussion on Figure 8, where we compare multi-agent and single-agent performances to extend the analysis on their limitations.

2. The authors understand and explain some of the common mistakes and failure modes of the single-agent approach. However, the discussion of the improvements made by the multi-agent

system lacks a similar depth. In particular, showing and discussing an exemplary case where the single-agent fails and the multi-agent succeeds would make a strong addition to the manuscript.

Author reply: We thank the reviewer for this suggestion. In the revision, we have added an example showing how a single-agent system frequently fails to coordinate chained tool calls when computing thermochemistry for a chemical reaction (Section S3, supporting information). To address this point, we have added a discussion on page 14-15 and Figure 7, which highlight why the single-agent approach fails in certain cases and how task decomposition in the multi-agent framework, where subtasks and context are isolated for each executor agent, enables multi-agent ChemGraph to achieve more reliable performance.

3. It is unclear where the key novelty lies in the ChemGraph approach

a) for the single-agent case: w.r.t. the integration of new tools in an existing single-agent approach like ChemCrow or CACTUS

b) for the multi-agent case: w.r.t. multi-agent systems like El Agente (Q), which the authors cite, and which is integrated with many of the same tools

Author reply: We thank the reviewer for raising this important point. To clarify the novelty of ChemGraph, we have substantially revised the section now titled “Distinctive Features of ChemGraph” (previously “Flexibility and Robustness of ChemGraph”) to explicitly compare against recent agentic workflows such as ChemCrow, CACTUS, and El Agente (Q). In particular:

While prior systems such as ChemCrow and CACTUS integrate individual tools with LLMs, ChemGraph introduces a highly modular architecture via ASE integration that unifies molecular simulation, property prediction, and structure manipulation within a single, extensible framework. This enables using a variety of computational chemistry packages without re-engineering the agent pipeline. Moreover, ChemGraph uniquely combines chemistry foundation models for fast yet accurate approximations with downstream high-fidelity simulations, allowing interactive exploration not supported in many other systems, yet.

Although systems like El Agente (Q) employ multiple agents, ChemGraph distinguishes itself through (i) systematic benchmarking across single- and multi-agent variants using standardized chemistry tasks with open-source release for reproducibility; (ii) cost-efficient and scalable design by deliberately incorporating small LLMs (e.g., GPT-4o-mini, Claude-3.5-haiku) and an open-weight LLM (Qwen-2.5-14B) to balance accuracy and resource constraints; and (iii) robustness analysis that probes the limits of coordination strategies, error handling, and tool integration under realistic scientific workloads, which are capabilities not explored in prior multi-agent frameworks.

Together, these features make ChemGraph an open, flexible, and resource-aware framework that is evaluated on a variety of computational chemistry and materials science tasks.

Reviewer #2 (Remarks to the Author):

Paper Review: ChemGraph: An Agentic Framework for Computational Chemistry Workflows

1. Recommendation

I believe this work introduces an interesting computational chemistry utility to the wider community, but the conversation around the benchmark is not very compelling and some details about the tools themselves are not very clearly articulated.

I'd accept this with Minor to Major Revisions.

2. Summary of the Paper

This paper highlights a new LLM-based tool: ChemGraph, which is a natural language interface to a set of computational chemistry tools through the LangGraph framework. The paper introduces the tool and discusses benchmarking results for a variety of tasks.

This paper makes no major applied scientific claims about their tool, but they discuss its impact in how it can utilize various LLM's to accomplish tasks to varying degrees of success to allow for a wider access to computational chemistry for those unfamiliar with programming or to accelerate computational chemistry research for those that are familiar.

They also discuss the differences in execution between single and multi-agent configurations of their tool.

Overall this is a paper that highlights the creation of an exciting new tool for the computational chemistry community.

3. Strengths

* Framework

I believe the authors do a good job of highlighting the structure of their agent / multi-agent workflow. It is clear how the tools are being used and the format which they are given.

* Diversity in Models

The model selection for this paper is a strong source of comparison to the current state of models. It is an interesting outlook to compare how certain models perform on similar tasks in a controlled setting like the benchmarks provided here.

* Application

The authors have introduced a tool that I could see serving in a useful way to the community. Lowering the barrier of entry to computational chemistry in this way could see a lot of benefits to computational design of molecules and materials.

* Code and Data Availability

The authors have been very transparent in their included code, data, and supporting information. This is a great mechanism to take, and greatly benefits the overall community with sharing approaches to integrating LLMs with scientific tooling.

4. Major Weaknesses / Areas for Major Revision

* Discussion on the Benchmarks

There was some discussion of the benchmark design but it did not go into much detail about the actual format of the questions asked. There is one example of a multi-turn interaction that gives a typical question, but it would be beneficial for an example to be provided for the major experiments. The benefit of an LLM is the natural language approach, so seeing what natural language is asked is helpful. I know there is some information in the codebase itself and the supporting information, but I believe there should be more in the main text.

Author reply: We thank the reviewer for this suggestion. To address this, we have expanded the manuscript to include more illustrative examples of natural language interactions with ChemGraph. Specifically, we have added a section titled “ChemGraph Workflow Demonstration” on Page 9 with Figure 3 to demonstrate a representative human-ChemGraph interaction in a single-agent setting. We have also expanded the “Multi-agent evaluations” section to include an example of a multi-agent interaction (Page 14-15 and Figure 7). We hope these examples provide clearer insight into the format and style of questions posed to ChemGraph, and how ChemGraph uses tool calls to solve the problems.

* The Benchmark

How were the number of instances determined? Why run some at higher instances than others? 30 instances seems insufficient for a sample size and then getting down to 10 for the more complicated questions where you'd like to see a more robust analysis. I understand with LLMs the API calls can be expensive, but do you believe that you are getting a large enough sample size for those sets of questions?

Author reply: We thank the reviewer for these questions. First, we want to point out that while there are a few benchmarks for agentic AI, there is no established benchmark for agentic workflows in computational chemistry/materials science. Recently published agentic AI for computational chemistry/material sciences typically focused on evaluation on only a few case studies. Accordingly, we have expanded the discussion at the

beginning of the “Benchmarking and Evaluation” section (Page 5), and in the updated “Distinctive Features of ChemGraph” section (Page 16-17).

We agree that our evaluation will benefit from larger sample sizes. Our original dataset of 260 instances was designed to be comparable to other widely used general agentic benchmarks, such as the Berkeley Function-Calling Leaderboard (BFCL) V3, in which subsets often contain about 100-200 question–answer pairs. In response to the reviewer’s feedback, we have expanded our evaluation dataset from 260 to 360 questions by increasing the number of instances from 15 to 30 instances across experiments 1-11. For the more complex tasks (*react2enthalpy* and *react2gibbs*), we expanded the evaluation from 10 to 15 instances. We note that for these two tasks, evaluations are repeated three times, including runs with the more expensive model (GPT-4o). As a result, we capped the number of instances at 15 to keep the total API cost manageable while still providing a robust assessment.

Also is there a sense of time in your benchmarks? How long does a response take to generate single or multi-agent? The computations are rather uncontrollable, but the time to perform a typical ReAct loop from start to end may be worth highlighting.

Author reply: We thank the reviewer for this question. We have added the benchmark timing in the supporting information section S4. We report the average end-to-end workflow time (LLMs + simulation) and the “Only Simulation” time as baseline. The difference between these two values reflects the ReAct loop overhead. For our benchmark, this overhead is less than one minute on average, with some exceptions for the Claude-3.5-haiku model.

We note that these timings should be interpreted as approximate, since LLM response time depends on several factors, including model latency at the time of benchmarking, the model’s ability to parallelize or batch tool calls, and the accuracy of tool-call generation. In future evaluation work, we plan to separate these factors more explicitly to provide a clearer picture of timing behavior across different models.

* Use of LLMs over standard software methods

I understand the benefit to using LLMs to serve as a natural language interface to individuals who may have little experience in programming, but I'd like to see a stronger reason why LLM-agents help to solve this problem in a better way over just a simple graphical user interface that taps into the same backend python wrappers that you have written for the LLM to utilize.

The discussion provided in the 'future directions' section should focus on the strengths of LLM-based agents over rigid GUI tools that can be designed to accomplish a similar goal.

Author reply: We thank the reviewer for this insightful comment and agree that this is a crucial point. The ideal path forward is not a matter of choosing between a Graphical User

Interface (GUI) and a Natural Language Interface (NLI), but rather in combining them to create a more powerful, intuitive, and flexible system. We have revised the "Limitations, safety and future directions" section to better articulate this vision.

While GUIs are excellent for lowering the entry barrier for standardized tasks, their primary limitation is rigidity. They typically offer pre-defined workflows, and incorporating novel methods or complex, multi-step procedures requires dedicated and often slow development cycles. This makes them less suited for the exploratory nature of scientific research.

LLM-based agents, like ChemGraph, address this limitation by enabling dynamic workflow composition. A key strength of our approach is that the agent can reason and sequence tools to accomplish tasks for which it was not explicitly pre-programmed. For instance, in our single-agent benchmark, the LLM autonomously determined the correct sequence of tool calls (`molecule_name_to_smiles`, `smiles_to_atomsdata`, and `run_ase`) to calculate a thermochemical property from just a molecule's name. This adaptability is crucial for novel research questions and scales to even more complex tasks, as demonstrated by our multi-agent system.

Ultimately, we envision a synergistic system where the NLI and GUI work together. A researcher could use natural language to describe a novel, complex workflow, and the GUI could dynamically populate with the corresponding tools and parameters, providing visual feedback. This hybrid approach would offer the best of both worlds: the flexibility and expressiveness of natural language combined with the intuitive, visual nature of a GUI. This integration makes the interface more natural and removes the steep learning curve often associated with feature-rich, GUI-only software. To underscore our commitment to this vision, our publicly available codebase already includes an experimental GUI for ChemGraph built with Streamlit, serving as an initial step toward this more integrated and user-friendly future. Accordingly, we revised the "Limitations, safety and future directions" section to clarify why LLM-based agents go beyond what GUI tools can offer by combining the benefits of both.

5. Minor Weaknesses / Suggestions for Minor Revision

* Grammar

There are a few sections that could use another pass on grammar checking to ensure clarity.

A few I noticed:

** Page 2: "that can perform molecular dynamics (MD) workflow" -> "workflows"

** Page 2: "In this work, we introduced" -> "we introduce"

Author reply: We have corrected the identified grammar issues and conducted a thorough grammar review of the entire manuscript.

* Computational Power

You mention where you sourced the LLM's from, either HPC or an API, but for actually running the workflow is there a requirement for compute power? Like is this something only a user of an HPC could run? Some discussion about the hardware used to perform the experiments would be beneficial. To clarify my question a little more, what is running the python code?

Author reply: We are thankful for the opportunity to clarify the computational cost for the evaluations. In the original version, all benchmarks were performed locally on a personal laptop. LLM inference was done through cloud APIs for the proprietary models (GPT-4o, GPT-4o-mini, Claude-3.5-haiku) and through ALCF endpoints for Qwen-2.5-14B. We note that Qwen-2.5-14B could also be run on a local machine, so high-performance computing (HPC) is not required for executing the benchmark workflow.

For the scientific tasks, we used efficient models such as machine learning potentials (MACE) and semi-empirical methods (xTB) and limited DFT calculations to small molecules. This kept the evaluation benchmark workflow computationally lightweight and suitable for modest hardware.

In the updated version, we reran the benchmarks (see supplementary information, section S4) on the Aurora supercomputer at Argonne only to obtain consistent timing measurements that were unaffected by local background processes. We emphasize that HPC resources are not necessary for the benchmark workflow, although ChemGraph can also be deployed on HPC systems if one is interested in running larger and more computationally demanding simulations.

* Multi-agent evaluation for GPT-4o

Maybe I missed it, but was there a reason there was no multi-agent run of Gpt-4o on the `react2enthalpy` and `react2gibbs` to compare with the other multi-agents?

Author reply: We thank the reviewer for pointing this out. In the initial submission, we excluded the multi-agent runs of GPT-4o on the react2enthalpy and react2gibbs tasks due to its lower throughput token limit and higher cost per token, which made repeated multi-agent evaluations less practical at the time. In the revised version, we have now performed these evaluations and included the results in Figure 8. Notably, multi-agent GPT-4o achieved perfect accuracy (100%) across three independent runs on both tasks, further reinforcing the effectiveness of the multi-agent design.

Reviewer #3 (Remarks to the Author):

The authors present ChemGraph based on LLM agents for autonomous computational chemistry simulations. They perform an important and honest benchmark of some typical tasks, and provide the data and code. I commend the authors for the nice work. It is a timely contribution, given the current trend of rapidly developing agentic tools for computational chemistry.

I only have a few rather technical comments:

1. Please provide the versions of the software used, i.e., Orca, etc, as they sometimes provide different results. Similar to the above, please provide the date of making benchmarks with some of the LLMs via API (i.e., for models which the authors cannot deploy locally, such as GPT-4 series). They might change with time, and we observed that some of these tools remember the previous requests through the API, if not explicitly turned off, which might influence the evaluations. That is, the evaluations might change if they are to be repeated.

Author reply: We thank the reviewer for this comment. We have added the specific versions of all software used, as well as the LLM tags for our evaluations, in the *Code Availability* section of the manuscript.

2. It would be interesting to see the time to solution for some of the typical tasks and, possibly, a breakdown of how much of the time is taken by agents and time taken by doing actual computational chemistry calculations.

Author reply: We thank the reviewer for this suggestion. In response, we reran our benchmarks to collect detailed timing data, which are now reported in Section S3 of the Supporting Information. On average, the agent processing time (excluding simulation) is under one minute, with the exception of a few cases involving the Claude-3.5-haiku model. The simulations themselves also average under one minute, ensuring that the benchmark remains both efficient and robust.

We note that these timings should be interpreted as approximate, since LLM response time depends on several factors, including model latency at the time of benchmarking, the model's ability to parallelize or batch tool calls, and the accuracy of tool-call generation. In future evaluation work, we plan to separate these factors more explicitly to provide a clearer picture of timing behavior across different models.

3. Could author elaborate more on how their work differs from the concurrent related works, i.e., El Agente (cited preprint, but it is now published in DOI 10.1016/j.matt.2025.102263), MatAgent (not cited: <https://doi.org/10.1002/mgea.70013>), Aitomia (not cited, arXiv:2505.08195), and DREAMS (appeared after ChemGraph's submission, arXiv:2507.14267)?

Author reply: We revised the section previously titled "Flexibility and Robustness of ChemGraph" to "Distinctive Features of ChemGraph" (Pages 16-17) to include discussion

of El Agente, MatAgent, Aitomia, and DREAMS, and to clarify how ChemGraph differs from these related works. We acknowledge that these studies have made important contributions in developing agentic AI systems with different designs for accelerating computational chemistry and materials discovery. Below is a summary of the distinctive features of ChemGraph.

While prior systems such as ChemCrow and CACTUS integrate individual tools with LLMs, ChemGraph introduces a highly modular architecture via ASE integration that unifies molecular simulation, property prediction, and structure manipulation within a single, extensible framework. This enables using a variety of computational chemistry packages without re-engineering the agent pipeline. Moreover, ChemGraph uniquely combines chemistry foundation models for fast yet accurate approximations with downstream high-fidelity simulations, allowing interactive exploration not supported in many other systems, yet.

Although systems like El Agente (Q) employ multiple agents, ChemGraph distinguishes itself through (i) systematic benchmarking across single- and multi-agent variants using standardized chemistry tasks with open-source release for reproducibility; (ii) cost-efficient and scalable design by deliberately incorporating small LLMs (e.g., GPT-4o-mini, Claude-3.5-haiku) and an open-weight LLM (Qwen-2.5-14B) to balance accuracy and resource constraints; and (iii) robustness analysis that probes the limits of coordination strategies, error handling, and tool integration under realistic scientific workloads, which are capabilities not explored in prior multi-agent frameworks.

Together, these features make ChemGraph an open, flexible, and resource-aware framework that is evaluated on a variety of computational chemistry and materials science tasks.

Reviewer #1 (Remarks to the Author):

I'm satisfied with the changes that the authors have made.

Author reply: We thank the reviewer for their helpful comments that improved the manuscript and their support for the publication.

Reviewer #2 (Remarks to the Author):

I believe that the authors have done a satisfactory job in responding to all 3 reviewers comments in a way that enhances the quality of the paper.

General Comments:

In reference to my comments, the inclusion of Figure 3 and Figure 7 provide a much more insightful view into the processes and capabilities of ChemGraph. The added discussion on a typical conversation and highlighting how the single-agent / multi-agents decide on their final response is a great benefit to the overall article.

The enhanced discussion on benchmarking and increase of the # of questions / instances is a good step in the right direction. I agree with the reasoning the authors have provided for why they chose the number they did, and I think the increase to 360 question-answer pairs satisfies any concern I had over not having a representative dataset to identify weaknesses in LLM responses.

The author's inclusion of the multi-agent GPT-4o run is great to see. The results gained from it's inclusion seem to have greatly supported the strengths of multi-agent task completion for this set of problems.

Points to address:

I believe all the points I had were addressed and no new major issues have appeared from the addition of the new content.

Author reply: We thank the reviewer for their helpful comments that improved the manuscript and their support for the publication.

Reviewer #3 (Remarks to the Author):

The revision has improved the manuscript and addressed some of the comments. However, it also introduced some new dubious claims and mistakes.

In the revised versions, the references are not complete; the paper cites 47 papers, while the References only list 35 papers.

Author reply: We apologize for the incomplete reference list. The references have now been corrected, and the manuscript has been updated accordingly.

In the following paragraph, they mention MatAgent twice but give different references to likely different publications: “MatAgent42 by Lv et al. is a single-agent system MatAgent44 by Takahara et al. relies on tool-augmented LLM reasoning (tested with GPT-4o and o3-mini) ...” Unfortunately, it is not possible to check where the mistake lies as the references 42 and 44 are not provided.

Author reply: We appreciate the reviewer for pointing this out. Indeed, there are currently two different agentic frameworks named "MatAgent" by different research groups. We added a short remark to prevent confusion for future readers.

In the distinctive features of ChemGraph, they mention flexibility and modularity due to the ASE use. They also emphasize several times the foundation model support, which they attribute in the response letter as ‘unique’. None of this is unique, as Aitomia is based on MLatom, which also enables integration of a wide range of simulation packages akin to ASE, and it also supports a host of foundation models (in fact, dozens of them). Another distinctive feature was said to be the use of small LLMs, but the previous work on Aitomia also explored them.

They also mention the ‘vision’ of combining ChemGraph with GUIs, but Aitomia also demonstrated the synergic benefits of combining agentic workflows with GUI. While the above independent work on Aitomia does not take away from the original contribution by the authors, the above statements must be revised to correctly reflect the current state of the art.

Author reply: We thank the reviewer for this comment and for pointing us to the relevant capabilities demonstrated in Aitomia. To address these concerns and more accurately reflect the current state of the art, we have made the following revisions:

1. **Section Renaming and Reframing:** We have renamed the section formerly titled "**Distinctive Features of ChemGraph**" to "**Key Features of ChemGraph**". This new title avoids overstating novelty and more appropriately emphasizes our design contributions in context. We have also added the following statement to this section:

“Within this evolving landscape, our goal is to highlight the key features of ChemGraph that contribute to this effort, without asserting exclusivity or novelty, but rather to contextualize our design choices and their impact.”

2. **Clarifying Foundation Model and Small LLM Claims:** After carefully reviewing the Aitomia preprint, we observed that the main implementation and deployment focus is on DeepSeek-V3, a large LLM, and that DeepSeek-r1:32B is used in a distinct role as a retrieval-augmented lookup tool rather than as the reasoning agent. A fine-tuned Qwen-7B model is also briefly mentioned in the Supporting Information, but it is demonstrated via only a single manually verified geometry optimization example.

In contrast, ChemGraph provides a systematic and reproducible benchmarking study of multiple small LLMs (e.g., Qwen2.5-14B, Claude 3.5 Haiku, GPT-4o-mini, etc.) across diverse computational chemistry tasks and directly compares these results with those from a large model (GPT-4o). This comprehensive evaluation of small LLMs, including their accuracy, reliability, runtime, and cost profiles, is, to our knowledge, not currently available in the literature. For this reason, we continue to present this aspect as a **key feature** of ChemGraph, while no longer using the term "distinctive."

3. **ASE vs MLatom for Modularity:** ChemGraph uses ASE to enable modular integration with a wide range of classical, semi-empirical, ML-based, and DFT simulation methods. ASE's broad compatibility and active community support make it a practical foundation for building reproducible and extensible agentic workflows. While we do not claim this as a unique capability and agree that MLatom provides a similar capability for Aitomia, we consider it a key feature of ChemGraph that contributes to its flexibility and ease of adoption. We have revised the text to reflect this perspective more clearly, without making exclusivity claims. We have also revised the sentence describing Aitomia to reflect their support for foundation models through MLatom: *“Aitomia⁴⁶ is a multi-agent system capable of performing computational tasks using a wide range of tools, from quantum mechanical methods to machine learning approaches, with DeepSeek-V3 as the core engine and many foundation models supported through the MLatom package.”*
4. **GUI Integration Clarification:** We would like to clarify that we did not present GUI integration as a distinctive or a novel feature of ChemGraph. The discussion on GUI was added in response to a direct request from another reviewer, who asked for a clearer explanation of how LLMs offer advantages beyond traditional GUI tools. Our intention was to emphasize the complementary nature of GUI and natural language interfaces, not to claim novelty over Aitomia or other related platforms.